# OpenReview forum: "SafeSora: Towards Safety Alignment of Text2Video Generation via a Human Preference Dataset"
_NeurIPS.cc/2024/Datasets_and_Benchmarks_Track — NeurIPS 2024 Track Datasets and Benchmarks Poster_

### Official Review · Reviewer_53du · 2024-07-26
**Helpfulness vs Harmlessness dataset for Text-to-Video models**

**Rating:** 6
**Confidence:** 5
**Correctness:** Yes, the claims seem to be correct.
**Clarity:** Yes

**Review:**

The paper has been written clearly and concisely. The dataset tackles an important problem in the text-to-video space.

**Strengths:**

- The dataset size is pretty large, making it useful for training models.
- Sub-categorization of helpfulness and harmlessness is well defined with correlation matrix between each category.
- Study of helpfulness vs harmlessness on existing open-source models gives more insights into the practical application of the models.

**Additional Feedback:**

Please read the opportunities for improvement section.

**Documentation:**

Yes

**Ethics:**

There are ethical concerns but are fairly addressed by the authors.

**Limitations:**

I would like the authors to add a discussion on bias mitigation in the dataset and list out potential biases that are present.

**Opportunities For Improvement:**

- Bias: The authors claim that allowing free perception of definitions of categories provide better annotations. However, this can create a bias in the dataset motivated by the demographics of the crowdworkers. This is mentioned slightly at L181, however no concrete information is given on it. Does SAFESORA have any bias on demographics (location, gender, age etc.) on the annotators? If yes, how do the authors mitigate it?

- Further, I will be interested in knowing other biases in the dataset?

- Definition of Correctness: In the correctness definition, it is mentioned that video should adhere to “natural laws”. Isn’t it ambiguous, what does natural laws mean here? Are these laws of physics? Or nature facts (like the sun rises in the east)?

- Fig 6(1) numbers: The figures shows that VC1 is better than VC2 in terms of following helpfulness-harmlessness tradeoff even though alignment increases. Do the authors have any insight of the reason behind this? A discussion on all four models based on their training dataset can be a good insight for future readers.

- Missing Correlation: The paper misses correlation between harmfulness vs harmlessness categories. L235 mentions contradiction between the two dimensions, however a detailed insight within each category would be interesting as well.

- Gap in motivation: The paper revolves around helpfulness vs harmlessness tradeoff in LLMs and analogically using in T2V models. However, there is a gap in introduction in motivating this correspondence. Adding more details around L33 would motivate the idea better. For eg. Why does such a correspondence exist?

**Relation To Prior Work:**

Yes

**Summary And Contributions:**

SAFESORA is a Text-video pair dataset that measures human preferences under helpfulness and harmlessness dimension. The dataset is crowdsourced and has detailed sub-categories under each dimension. The authors release a large harmfulness vs harmlessness dataset for future text-to-video models.

---

> ### Author Rebuttal · Authors · 2024-08-17
>
> > **OFI1:** Bias: The authors claim that allowing free perception of definitions of categories provide better annotations. However, this can create a bias in the dataset motivated by the demographics of the crowdworkers. This is mentioned slightly at L181, however no concrete information is given on it. Does SAFESORA have any bias on demographics (location, gender, age etc.) on the annotators? If yes, how do the authors mitigate it?
> >
> > **L1:** I would like the authors to add a discussion on bias mitigation in the dataset and list out potential biases that are present.
>
> **Reply to OFI1,L1:** We appreciate the opportunity to discuss with you about the bias of demographics.
>
>
> Regarding the rationale for allowing crowdworkers a certain degree of subjective judgment:
>
> - Firstly, alignment research inherently aims to align with a group’s preferences, which are **naturally subjective** and therefore unavoidably carry some group-specific biases.
>
> - Secondly, the primary objective of our research is **to provide a foundational dataset for studying human preferences and validating alignment algorithms**, rather than creating a dataset that can align with any arbitrary group. Therefore, we tend to allow more freedom for crowdworkers to better express genuine preferences.
>
> For researchers with specific alignment needs that differ from the groups represented in our data, it may be necessary to conduct additional annotations tailored to their target group. **We hope that our proposed annotation process and detailed documentation will be of some assistance in this regard.**
>
> This attribute indeed proves useful for the application of the SafeSora dataset, and we will clarify it in the latest manuscript:
>
> **Demographic attributes:** We partnered with a local data labeling service provider.Our crowdworkers have a gender distribution of 62.5% female and 37.5% male. Age-wise, 52.5% are aged 20-30, 40% are 30-40, 7.5% are others. All participants belong to the Asian ethnic group and are rooted in the East Asian cultural sphere. Educationally, 65% hold a Bachelor's degree and 35% have a technical/vocational diploma.
>
> We do not overlook the issue of bias. We have implemented a standardized protocol [1] to limit subjective behavior and reduce bias, including:
>
> - **Standardized Procedures:** We guide crowdworkers in structured thinking through a two-phase annotation process, ensure quality through a secondary review mechanism, and align the secondary review standards through regular communication between the research and annotation teams.
>
> - **Comprehensive Annotation Documentation:** Particularly in safety-related aspects, we have referenced [2][3] to ensure that definitions in the annotation documentation are relatively objective and clear.
>
> - **Decoupled Annotation Dimensions:** We mitigate subjective trade-offs between different dimensions by decoupling helpfulness and harmlessness and by setting sub-dimensions.
>
> **References：**
>
> [1]Mikołajczyk-Bareła et al. A survey on bias in machine learning research. arXiv preprint arXiv:2308.11254.
>
> [2]Motion Picture Association film rating system https://www.filmratings.com/downloads/rating_rules.pdf
>
> [3]GARM: Brand Safety Floor + Suitability Framework https://wfanet.org/knowledge/item/2022/06/17/GARM-Brand-Safety-Floor--Suitability-Framework-3
>
> ---
>
> > **OFI2:** Further, I will be interested in knowing other biases in the dataset?
>
> **Reply to OFI2:** We have observed some annotation differences arising from legal or cultural factors.
>
> For example, as illustrated in Figure 1 of the paper, the left image shows harmful classification of text prompts using GPT-4, while the right image displays the results of manual video annotations. It is evident that the proportion of the **contraband** category is significantly higher in the manual annotations. This discrepancy arises because our crowdworkers' judgments about prohibited items differ from those of GPT-4. For instance, civilian possession of firearms is annotated as harmful contraband by our crowdworkers (due to local legal restrictions), whereas GPT-4 considers it harmless.
>
> ---
>
> > **OFI3:** Definition of Correctness: In the correctness definition, it is mentioned that video should adhere to “natural laws”. Isn’t it ambiguous, what does natural laws mean here? Are these laws of physics? Or nature facts (like the sun rises in the east)?
>
> **Reply to OFI3:** Thank you for your suggestion. We will revise this section to improve clarity. **Both of the physical laws and natural facts you mentioned are indeed aspects we consider in this dimension.** Similar details are provided in our discussion of correctness (see Appendix D.1.2). Based on your feedback, we will update the main text as follows:
>
> **Correctness:** Evaluate whether objects in the video have correct shapes and movements. Unless explicitly stated in the instructions, objects in the video should have attributes such as shape, color, and size that align with natural facts. Their movements should follow physical laws, such as gravity and conservation of momentum.

---

> > ### Author Rebuttal · Authors · 2024-08-17
> >
> > > **OFI4:** Fig 6(1) numbers: The figures shows that VC1 is better than VC2 in terms of following helpfulness-harmlessness tradeoff even though alignment increases. Do the authors have any insight of the reason behind this? A discussion on all four models based on their training dataset can be a good insight for future readers.
> >
> > **Reply to OFI4:** In Figure 6(1), VC1 (VideoCrafter1) generated fewer harmful T-V (Text-Video) pairs compared to VC2 (VideoCrafter2). In fact, by directly observing their video generation results, we found that **the higher proportion of harmless outputs from VC1 is due to its lower generation capability**. For many prompts in the evaluation set, VC1 failed to generate content that correctly matches the instructions. Specifically, for many safety-critical prompts, VC1 generated invalid videos, which were classified as safe.
> >
> > For all the models we evaluated, since most open-source video generation models haven't undergone alignment training, **the stronger the model's capability, the more likely it is to generate harmful content according to the text instructions, resulting in lower harmlessness**. Therefore, in Figures 6(2) and (3), we see an inverse trend between harmlessness and helpfulness across the four models.
> >
> > ---
> >
> > > **OFI5:** Missing Correlation: The paper misses correlation between harmfulness vs harmlessness categories. L235 mentions contradiction between the two dimensions, however a detailed insight within each category would be interesting as well.
> >
> > **Reply to OFI5:**  Thank you for your reminder. However, there seems to be a misunderstanding that needs clarification. The harmfulness partial order and harmlessness classification are two different types of annotations, targeting T-V-V pairs and T-V pairs, respectively. Therefore, it is difficult to calculate the correlation between them.
> >
> > On the other hand, L235 mentions the conflict between the two main dimensions of helpfulness and harmlessness. **Detailed analyses within each dimension have actually been provided earlier in the text**:
> >
> > - **Harmlessness:** Correlation analysis between harmful class labels (L205) and consistent treatment of harmful class labels with harmlessness preferences (L216);
> > - **Helpfulness:** Correlation analysis between helpfulness sub-dimensions and the main dimension (L220).
> >
> > ---
> >
> > > **OFI6:** Gap in motivation: The paper revolves around helpfulness vs harmlessness tradeoff in LLMs and analogically using in T2V models. However, there is a gap in introduction in motivating this correspondence. Adding more details around L33 would motivate the idea better. For eg. Why does such a correspondence exist?
> >
> > Thank you for your constructive feedback. We will further elaborate on the conflict between helpfulness and harmlessness, as well as the significance of decoupling these two dimensions.
> >
> > For the same comparison pair generated by a safety-critical prompt, helpfulness and harmlessness often yield opposite preferences. This phenomenon is reflected in our dataset analysis (L235). The reason is that **a higher helpfulness score indicates a response that better meets the prompt's requirements, which in the case of safety-critical prompts, also increases the likelihood of generating harmful videos.**
> >
> > To mitigate this conflict, using a decoupled annotation approach can **prevent annotators from having to subjectively judge the trade-off between the two dimensions**, thereby improving the quality of data collection to some extent. Additionally, decoupled annotations provide distinct insight for different dimensions, **allowing researchers to adjust the weight of each dimension during training**.
> >
> > Based on your suggestions, we will discuss these contents more in the instruction section of the paper.
> >
> > ---
> >
> > Finally, we express our sincere gratitude for the insightful questions and suggestions, which have been helpful to our work and will be integral to our revised manuscript.
> > **If our efforts can address your concerns, we sincerely hope you will recognize our work.**

---

> > ### Comment · Reviewer_53du · 2024-08-30
> > **Conflicting opinion on subjectivity of annotations**
> >
> > Thank you authors for your response and answering my questions. Unfortunately, I disagree with author's suggestions that subjectivity of choice in understanding harmfulness (and natural laws) is a good decision. It conflicts with the paper's motivation in clearly defining harm categories and even author's first and second response conflict in their stance on this issue.
> >
> > Due to these reasons, I am decreasing my score to reject (4).

---

> > > ### Author Response · Authors · 2024-08-31
> > > **Clarification of Serious Misunderstanding**
> > >
> > > We would like to clarify that our emphasis has consistently been **on a certain degree of subjectivity, not on unrestrained subjectivity**. Thus, there is **no conflict between the subjectivity and our motivations**, particularly in light of the harmfulness and natural laws you mentioned, which are designed to limit harmful biases. Our reasons are as follows:
> > >
> > > - **Fact 1: Human preferences inherently possess subjectivity** — The goal of alignment tasks is to align the model's distribution with the preferences of a particular group. These preferences, such as the group's likes or dislikes, are inherently subjective expressions [1].
> > > - **Fact 2: Unconstrained subjectivity can introduce harmful biases** — Due to the boundaries set by local laws, universal ethical standards, and natural laws, annotation should not cross these boundaries and introduce harmful biases. For instance, breaking the law to perform a task is harmful.
> > > - **Inference: The data collection for alignment tasks requires a certain degree of subjectivity, but must constrain the expression of harmful biases** — It is essential to determine which dimensions can tolerate a certain degree of subjectivity and which aspects require constraints. This can be achieved by establishing processes and guidelines to avoid the introduction of harmful biases.
> > > - **Technical Solution 1: Decoupling across different dimensions** — The degree of permissible subjectivity varies across different dimensions. For example, harmlessness requires stricter constraints compared to helpfulness. Therefore, human preferences need to be decoupled to better guide subjective expressions across different dimensions. We divided the annotation process into two major dimensions: helpfulness and harmlessness, along with four sub-dimensions under helpfulness.
> > > - **Technical Solution 2: Limited subjective expression based on annotation guidelines** — In the annotation process, we require crowdsourcing workers to first adhere to the detailed annotation documents. Our annotation guidelines effectively reflect the boundaries in different dimensions, imposing a certain degree of constraint on subjective expressions to avoid generating harmful biases. For example, we have developed detailed explanations for twelve harmful categories based on local laws and authoritative documents precisely because violations of these would lead to harmful biases.
> > >
> > > In summary, it is a misunderstanding to broadly describe subjectivity as being in conflict with our work; such a characterization is a heartbreaking cover-up of our efforts. In fact, the solution to the subjectivity and objectivity issues in specific sub-dimensions like harmlessness and natural rules, which you mentioned, is **one of the technical contributions** of our work.
> > >
> > > Despite receiving your feedback on the last day, we sincerely hope to continue this discussion. **I genuinely hope you will re-evaluate our paper and response and provide a new assessment.**
> > >
> > > ---
> > >
> > > **Reference:**
> > >
> > > [1] Yuntao Bai, et al. "Training a Helpful and Harmless Assistant with Reinforcement Learning from Human Feedback"

---

> ### Author Response · Authors · 2024-08-23
> **Hope to Get Your Reply**
>
> Dear Reviewer 53du,
>
> As the deadline is nearing, we wanted to gently follow up on our recent submission. Your feedback is highly valuable to us, and we would appreciate any updates or further guidance you might have regarding our revisions and responses.
>
> Thank you for your time and consideration.

---

### Official Review · Reviewer_XZhL · 2024-07-28
**SafeSora**

**Rating:** 7
**Confidence:** 4
**Correctness:** See above
**Clarity:** See above

**Review:**

- The paper is well-written and motivated with clarity over technical details
- SafeSora dataset is novel and differs from existing works by focusing on preferences across text-video pairs
- The work includes preference annotations across sub-dimensions of harmfulness and helpfulness while providing detailed correlation studies and perspectives.
- The AI feedback and human feedback agreement analysis showcases importance of human feedback for this task
-The application of SafeSora as a content moderation training set is interesting and impactful for future research
- The authors also train 2 preference aligned models to show the usefulness of the preferences in the SafeSora dataset
- The work lacks inter annotator agreement and quality control statistics about the generated dataset

**Strengths:**

- SafeSora dataset is novel and differs from existing works by focusing on preferences across text-video pairs, the size of the dataset is good and can provide rich annotations for training. alignment etc.
- The detailed sub-dimensions of helpfulness and harmlessness is interesting and addresses an important point of subjectivity during preference annotations. The associated correlation study can be useful for future alignment research across modalities
- The AI feedback and human feedback agreement analysis showcases importance of human feedback for this task
-The application of SafeSora as a content moderation training set is interesting and impactful for future research
- The authors also train 2 preference aligned models to show the usefulness of the preferences in the SafeSora dataset. Would it be possible to train another more advanced alignment algorithm like DPO ?

**Additional Feedback:**

N/A

**Documentation:**

URL provided and IRB documentation on website

**Limitations:**

Yes. authors have identified the societal impact fo their work and justified the technical limitations of their work. Additionally, the authors have disclosed the fair use concerns, copyright license and associated IRB for the human annotation.

**Opportunities For Improvement:**

- I suggest the authors add an example of the dataset/task in the main text of the paper
- What were the inter-annotator agreement from the crowdworkers for the entire dataset ? I would like to see how disagreement were handled during annotation and how many annotations per data point was collected.
- What are the qualifications of the secondary quality control department ?
- The authors mention that the annotations are subjective due to various demographic attributes of the crowdworkers. Is it possible to analyze the variable of labels/preferences across various demographic attributes like culture, ethnicity or educational background ?

**Relation To Prior Work:**

I encourgae the authors to include alignment research in the related work section of the main text

**Summary And Contributions:**

This work introduces, SafeSora, a human preference dataset for text-to-video tasks. SafeSora focuses on the aspects of harmlessness and helpfulness in Text-Video pairs and collect fine-grained preferences across various subdimensions. The dataset has value and ill be useful to the community. The nuanced preferences across various subdimensions and associated copyright licenses will make this work widely distributed and useful.

---

> ### Author Rebuttal · Authors · 2024-08-17
>
> > **S4:** The authors also train 2 preference aligned models to show the usefulness of the preferences in the SafeSora dataset. Would it be possible to train another more advanced alignment algorithm like DPO ?
>
>
> As outlined in the Future Work section, we plan to develop more advanced alignment algorithms. Here, we have trained a model using the DPO algorithm and compared its performance with the baseline alignment algorithm mentioned in the paper, evaluating both on the SafeSora dataset.
>
> **Implementation.** We implemented our own DPO algorithm based on the formulas provided in the paper [1]. Since the official open-source code focuses on the text-to-image generation, we adapted their code to create a text-to-video version.
>
> **Results.** We trained our model on a dataset subset using both the BoN and DPO algorithms. Post-training, we evaluated the models on a separate dataset, calculating win rates with our reward model. The DPO-trained model outperformed the pre-trained model by 6% and the BoN-trained model by 2%.
>
> These results provide a **preliminary indication** of the effectiveness of DPO training. However, there are several challenges, such as DPO requires the preferred one as good as expert one in the preference pair. Therefore, it is not optimal to train DPOs directly with SafeSora without special filtering.  We will **continue to develope** a more efficient and effective alignment algorithm that fully leverages our dataset.
>
> [1] Wallace et al, 2024. Diffusion model alignment using direct preference optimization.
>
> ---
>
> > **OFI1:** I suggest the authors add an example of the dataset/task in the main text of the paper
>
> Thank you for your suggestion. We also realized that including an example of the dataset in the main text would help readers better understand our work. We will add the example as shown in **attached figure** to Section 3 of the latest manuscript.
>
> ---
>
> > **OFI2:** What were the inter-annotator agreement from the crowdworkers for the entire dataset ? I would like to see how disagreement were handled during annotation and how many annotations per data point was collected.
>
> **Reply to OFI2:** We calculated the inter-annotator agreement for the entire dataset, which resulted in a rate of 85.97%.
>
> Due to budget and time constraints, we did not conduct large-scale duplicate annotations. Our process primarily involved two rounds of annotations, with a third round conducted when there was a disagreement between the first two. In our dataset, 68.86% of the annotated items underwent two rounds of annotation, while 31.14% received a third round. The final outcome was determined by a simple majority vote. Additionally, any data batches that failed to pass the QC department or the researchers' spot checks were re-annotated.
>
> ---
>
> > **OFI3:** What are the qualifications of the secondary quality control department ?
>
> **Reply to OFI3:** The Quality Control (QC) department is composed of **full-time professionals from our partnering data annotation service provider**. Their qualifications are ensured through the company's internal training and certification processes, and they possess extensive experience and expertise in data annotation. As part of our data review process, the QC department examines the entire dataset to ensure compliance with the annotation guidelines, in contrast to the researchers' spot-checks, which cover 20% of the data. Our research team also engages in bi-weekly meetings with the QC department to ensure they fully understand our annotation requirements and can effectively communicate this feedback to the annotation team.
>
> ---
>
> > **OFI4:** The authors mention that the annotations are subjective due to various demographic attributes of the crowdworkers. Is it possible to analyze the variable of labels/preferences across various demographic attributes like culture, ethnicity or educational background ?
>
> **Reply to OFI4:** It identifies a highly meaningful research direction—analyzing the relationship between the demographic attributes of aligned groups and their varying preferences. Logically, such an analysis could be beneficial during alignment training, especially when aligning multiple groups or when the attributes of the aligned groups change. It is an intriguing possibility that studying group attributes might reduce the cost of aligning with a new group, though definitive conclusions will require further experiments and data.
>
> To support related research, we will explicitly include the following demographic attributes of our crowdworkers in the latest manuscript:
>
> **Demographic attributes:** We partnered with a local data labeling service provider. Our crowdworkers have a gender distribution of 62.5% female and 37.5% male. Age-wise, 52.5% are aged 20-30, 40% are 30-40, 7.5% are others. All participants belong to the Asian ethnic group and are rooted in the East Asian cultural sphere. Educationally, 65% hold a Bachelor's degree and 35% have a technical/vocational diploma.
>
> Unfortunately, it is challenging to analyze the impact of different group attributes based solely on the SafeSora data. On the one hand, this research focus was not part of our initial considerations, so we did not organize the Data Labeling Services by segmenting crowdworkers into different groups based on demographic attributes. On the other hand, due to privacy concerns, our partnering professional data annotation service provider has declined to provide individual-level data, making it impossible to disaggregate the results.
>
> Thank you very much for suggesting this valuable research direction. We are keen to include this analysis in our future work.
>
> ---
>
> Finally, we express our sincere gratitude for the insightful questions and suggestions, which have been helpful to our work and will be integral to our revised manuscript.
> **If our efforts can address your concerns, we sincerely hope you will recognize our work.**

---

> > ### Comment · Reviewer_XZhL · 2024-08-19
> > **Response to rebuttal**
> >
> > I thank the authors for their detailed replies to my concerns. All my concerns are adequately addressed and I am increasing my score. Goodluck!

---

> > > ### Author Response · Authors · 2024-08-20
> > > **Thank Reviewer XZhL for Approving Our Work**
> > >
> > > We sincerely appreciate your acknowledgment and are deeply encouraged by your decision to raise our rating. It is our honor to address your concerns, which have been helpful to our work and will be integral to the improvements in our final version.

---

### Official Review · Reviewer_k4id · 2024-07-31
**Important dataset with good analysis and documentation**

**Rating:** 8
**Confidence:** 5
**Correctness:** Methods and analysis are correct.

**Review:**

The dataset created by the authors is important and merits acceptance. There are a few suggestions I have given in the "Opportunities for Improvement" but overall, this paper is well-written and is of sufficient importance to be accepted.

**Strengths:**

- Large dataset with a diverse set of prompts
- Multiple generations per prompt
- Multiple video generation models
- Human and AI evaluations
- Evaluations across multiple important modalities (harmless and helpfulness) with multiple sub-components of these modalities
- Baselines showing the utility of this dataset for evaluation and fine-tuning (alignment).

**Additional Feedback:**

Minor issues or typos:
- In line 108, not sure what "were generated using large language models" means. I think you mean that 42.30% of the prompts were ehanced by an LLM prior to video generation?

**Clarity:**

Writing is detailed and clear. Some minor points are mentioned in "Additional Feedback".

**Documentation:**

Well documented. Dataset and code are available online. Some suggestions are included in the "Opportunities for Improvement."

**Ethics:**

Ethical issues are addressed, and I find no serious ethical concerns here.

**Limitations:**

Limitations are adequately discussed.

**Opportunities For Improvement:**

- I suggest including the source of the prompt (from online dataset or from the researchers) in the released datasets, as well as when the prompt is modified by AI.
- For evaluations, I wonder whether there are objective metrics that are useful in addition to the human and/or AI annotations. For example, for assessing "Informativeness," are there simpile metrics that could be used (e.g., something like how much entropy in the sequence of frames given the first frame)? Or for assessing "Aesthetics," can something like an average across frames of a standard image quality metric be used? If possible, I would encourage the authors to consider and include such metrics as it increases the utility of their dataset, especially when such metrics are found to align well with human annotations (as these metrics and generally much faster and cheaper to compute, as well as potentially easier to include in any kind of optimization procedure).

**Relation To Prior Work:**

Prior work is adequately discussed.

**Summary And Contributions:**

The authors create a video preference dataset for AI-generated videos. Their dataset is the first of its kind and is comprehensive in its evaluations. This is an important piece of work for alignment and safety of video generation.

---

> ### Author Rebuttal · Authors · 2024-08-17
>
> > **OFI1:** I suggest including the source of the prompt (from online dataset or from the researchers) in the released datasets, as well as when the prompt is modified by AI.
>
> **Reply to OFI1:** Thank you for your suggestion. We have organized the sources of all the prompts and integrated them into our dataset. For details, please refer to the link: https://huggingface.co/datasets/PKU-Alignment/SafeSora-Prompt.
>
> In this prompt dataset, the sources of the prompts are categorized into six groups:
> - **two categories for real user prompts**, Discord and VidProM;
> - **three categories for prompts constructed** using images and those specifically designed based on harmful content classifications, midjourney-detailed-prompts, LAION-400M, and HarmCategory;
> - **one category for prompts refined using LLMs**, AI-refined. The AI-refined prompts are saved along with the UID of the original prompt.
>
> ---
>
> > **OFI2:** For evaluations, I wonder whether there are objective metrics that are useful in addition to the human and/or AI annotations. For example, for assessing "Informativeness," are there simpile metrics that could be used (e.g., something like how much entropy in the sequence of frames given the first frame)? Or for assessing "Aesthetics," can something like an average across frames of a standard image quality metric be used? If possible, I would encourage the authors to consider and include such metrics as it increases the utility of their dataset, especially when such metrics are found to align well with human annotations (as these metrics and generally much faster and cheaper to compute, as well as potentially easier to include in any kind of optimization procedure).
>
> This is a valuable suggestion, and we will incorporate some traditional evaluation methods into the SafeSora library. While these simpler methods may not precisely capture human value preferences, they typically offer faster computation and lower costs. We hope these methods will complement feedback-based approaches and allow for comparative analysis. For specific implementation details, please refer to the link: https://github.com/PKU-Alignment/safe-sora/pull/5.
>
> In the SafeSora library, we will introduce four traditional methods to compare with a specific preference in SafeSora. These methods include:
>
> - **PSNR vs. Informativeness (agreement ratio - 0.732):** PSNR measures the quality of a reconstructed or compressed image/video by comparing it to the original, assessing their similarity. By analyzing PSNR between the first and subsequent frames, we evaluate dynamic changes in the video.
> - **HPSv2 vs. Aesthetic (agreement ratio - 0.603):** HPSv2 predicts human preferences for image beauty. We use it to assess the aesthetics of each video frame, averaging the results of frames for an overall aesthetic measure.
> - **CLIP vs. Instruction Following (agreement ratio - 0.579):** CLIP assesses instruction adherence by evaluating the similarity between the prompt and video frames and averaging the results of frames for an overall instruction-following measure.
>
> Regarding correctness, it is challenging to find a traditional method for evaluation. This is because this dimension relies on knowledge, so for now, we can only rely on evaluations based on large models. **More methods will be added gradually in the future.**
>
> ---
>
> > **AF1:** Minor issues or typos: In line 108, not sure what "were generated using large language models" means. I think you mean that 42.30% of the prompts were ehanced by an LLM prior to video generation?
>
> **Reply to AF1:** Thank you for highlighting this issue. The description may indeed be ambiguous. Our intention was to convey that 42.30% of the prompts were enhanced using LLMs before being used for video generation. We will clarify this in the updated manuscript.
>
> ---
>
> Finally, we express our sincere gratitude for the insightful questions and suggestions, which have been helpful to our work and will be integral to our revised manuscript.
> **If our efforts can address your concerns, we sincerely hope you will recognize our work.**

---

> ### Author Response · Authors · 2024-08-23
> **Hope to Get Your Reply**
>
> Dear Reviewer k4id,
>
> As the deadline is nearing, we wanted to gently follow up on our recent submission. Your feedback is highly valuable to us, and we would appreciate any updates or further guidance you might have regarding our revisions and responses.
>
> Thank you for your time and consideration.

---

### Official Review · Reviewer_3Hpk · 2024-08-01
**A timely dataset contribution for video generative models**

**Rating:** 7
**Confidence:** 4

**Review:**

The paper is well-written and structured, making it easy to follow. The figures and tables are clear and effectively illustrate the key points. The supplementary materials, including appendices, provide additional details that enhance the understanding of the dataset and methodologies used. This paper presents a significant and valuable contribution to the field of AI safety and alignment. As far as I am aware this is the first video generation allignment dataset, coupled with a thorough analysis and demonstration of its applications, it provides a solid foundation for future research in aligning text-to-video generation with human values. The authors' emphasis on ethical considerations and limitations further strengthens the paper. I strongly recommend its acceptance for publication.

**Strengths:**

1. Novelty and Importance: The SAFESORA dataset addresses a significant gap in the AI community by providing a comprehensive dataset focused on aligning text-to-video generation with human values. This is an important step forward in mitigating the risks associated with harmful outputs from LVMs.

2. Detailed Annotation Process: The authors provide a thorough description of the two-stage annotation process, which includes subdividing helpfulness into four sub-dimensions and harmlessness into 12 sub-categories. This structured approach ensures high-quality annotations and captures in-depth human preferences.

3. Robust Methodology: The methodology is well-defined and includes detailed steps for prompt collection, video generation, and human annotation. The use of LLMs like GPT-4 for prompt refinement and preliminary classification adds a layer of sophistication to the process.

4. Comprehensive Analysis: The analysis section offers insightful findings on the correlations between different dimensions of human preferences, highlighting the distinctiveness of harm categories and the tension between helpfulness and harmlessness.

5. Applications and Future Research: The authors effectively demonstrate the utility of the SAFESORA dataset through applications such as training a text-video moderation model and fine-tuning LVMs. They also propose potential future research directions, which adds value to the paper.

6. Ethical Considerations: The paper addresses ethical considerations, including the potential negative societal impacts of the dataset and the importance of responsible use. This demonstrates the authors' commitment to ethical AI research.

**Additional Feedback:**

No additional feedback

**Clarity:**

The paper is well-written and structured, making it easy to follow. The figures and tables are clear and effectively illustrate the key points. The supplementary materials, including appendices, provide additional details that enhance the understanding of the dataset and methodologies used.

**Correctness:**

The work and conclusions it draws appear correct to me. I believe the dateset construction and evaluation is sound.

**Documentation:**

The submission is well-documented with appropriate links and dataset cards.

**Ethics:**

I do not see ethical concerns. The submission includes details of human data collection and appropriate IRB form.

**Limitations:**

1. Limited Generalizability: While the dataset is comprehensive, the authors acknowledge that it is impossible to cover all scenarios. The dataset should be expanded over time to include a broader range of prompts and potential harmful outputs.

2. Baseline Algorithm Efficiency: The baseline algorithms provided are primarily for validating the dataset's effectiveness and may not be sufficiently efficient for alignment purposes. Future work should focus on developing more efficient alignment algorithms.

3. Human Feedback vs. AI Feedback: The comparison between human feedback and AI feedback shows a low agreement in preference assessments. This highlights the current limitations of multi-modal LLMs in achieving consensus with human annotations and suggests the need for further research in this area.

**Opportunities For Improvement:**

1. Dataset Expansion: The authors should consider expanding the dataset to cover more diverse scenarios and potential harmful outputs. This will enhance the generalizability of the findings and applications.
2. Algorithm Development: Future work should focus on developing more efficient and robust alignment algorithms to utilize the SAFESORA dataset effectively.
3. Enhanced AI Feedback Analysis: Additional research is needed to improve the agreement between human feedback and AI feedback, potentially by developing more advanced multi-modal LLMs capable of processing video input.

**Relation To Prior Work:**

Yes, as far as I know this is the first dataset for alignment of video generation models of this scale.

**Summary And Contributions:**

The authors introduce SAFESORA, a novel dataset designed to align text-to-video (T2V) generation with human values by capturing human preferences in T2V tasks along two primary dimensions: helpfulness and harmlessness. The dataset includes 14,711 unique text prompts, 57,333 videos generated by four distinct large vision models (LVMs), and 51,691 pairs of preference annotations labeled by humans. The authors demonstrate the utility of the dataset through several applications, including training a text-video moderation model and aligning LVMs with human preferences.

---

> ### Author Rebuttal · Authors · 2024-08-17
>
> > **OFI1:** Dataset Expansion: The authors should consider expanding the dataset to cover more diverse scenarios and potential harmful outputs. This will enhance the generalizability of the findings and applications.
> >
> > **L1:** Limited Generalizability: While the dataset is comprehensive, the authors acknowledge that it is impossible to cover all scenarios. The dataset should be expanded over time to include a broader range of prompts and potential harmful outputs.
>
> **Reply to OFI1,L1:** Thank you for suggesting an expansion of the dataset. We are currently in the process of collecting new data and anticipate releasing it as open-source by the end of next month at the latest. Our expansion efforts are concentrated in two key areas:
>
> - **New Models**: Since the submission of our first paper, the community has open-sourced several outstanding text-to-video generation models, such as Open-Sora and Open-Sora-Plan. Therefore, one of our directions for dataset expansion is to include the generated outputs of these new models into our dataset.
>
> - **New Text Prompts**: Through red-teaming exercises and continuous effort, we have gathered new text prompts. By using these prompts to generate additional videos, we aim to broaden the coverage of our dataset across a wider range of scenarios.
>
> ---
>
> > **OFI2:** Algorithm Development: Future work should focus on developing more efficient and robust alignment algorithms to utilize the SAFESORA dataset effectively.
> >
> > **L2:** Baseline Algorithm Efficiency: The baseline algorithms provided are primarily for validating the dataset's effectiveness and may not be sufficiently efficient for alignment purposes. Future work should focus on developing more efficient alignment algorithms.
>
> As outlined in the Future Work section, we plan to develop more advanced alignment algorithms. Here, we have trained a model using the DPO algorithm and compared its performance with the baseline alignment algorithm mentioned in the paper, evaluating both on the SafeSora dataset.
>
> **Implementation.** We implemented our own DPO algorithm based on the formulas provided in the paper [1]. Since the official open-source code focuses on the text-to-image generation, we adapted their code to create a text-to-video version.
>
> **Results.** We trained our model on a dataset subset using both the BoN and DPO algorithms. Post-training, we evaluated the models on a separate dataset, calculating win rates with our reward model. The DPO-trained model outperformed the pre-trained model by 6% and the BoN-trained model by 2%.
>
> These results provide a preliminary indication of the effectiveness of DPO training. However, there are several challenges, such as DPO requires the preferred one as good as expert one in the preference pair. Therefore, it is not optimal to train DPOs directly with SafeSora without special filtering.  We will **continue to develope** a more efficient and effective alignment algorithm that fully leverages our dataset.
>
> [1] Wallace et al, 2024. Diffusion model alignment using direct preference optimization.
>
> ---
>
> > **OFI3:** Enhanced AI Feedback Analysis: Additional research is needed to improve the agreement between human feedback and AI feedback, potentially by developing more advanced multi-modal LLMs capable of processing video input.
> >
> > **L3:** Human Feedback vs. AI Feedback: The comparison between human feedback and AI feedback shows a low agreement in preference assessments. This highlights the current limitations of multi-modal LLMs in achieving consensus with human annotations and suggests the need for further research in this area.
>
> It highlights a valuable research direction: enhancing the effectiveness of general multimodal large language models in text-to-video annotation tasks. Here, we add an experiment related to this research, comparing **the number of extracted frames as input**, **different multimodal models**, and the effect of **using CoT (Chain of Thought)**. The results of the experiment are shown in the **attached PDF table**.
>
> We observed the following phenomena:
>
> - **The agreement between multimodal large language models and human remains low** in the preference of text-to-video generation task. According to the data in the table, most consistencies are below 0.7.
> - **Increasing the number of frames actually reduces the annotation effectiveness.** This could be because current multimodal large models are primarily geared towards solving image tasks, and their ability to understand videos is still lacking.
> - GPT-4o-Mini and GPT-4o-0806 have enhanced their safety mechanisms, resulting in many instances of **refusing to answer**, which leads to decreased performance in the annotation task.
> - **CoT does not significantly improve performance in this task.** As shown in the table, the consistency of CoT is lower than that of the zero-shot method. The CoT approach we used involves first analyzing relevant content of a single video, then comparing the videos to give a preference. However, the multimodal large models began to show biases during the first step of analysis, leading to a decrease in the final consistency.
>
> The results and analysis above will be integrated into our latest manuscript.
>
> ---
>
> Finally, we express our sincere gratitude for the insightful questions and suggestions, which have been helpful to our work and will be integral to our revised manuscript.
> **If our efforts can address your concerns, we sincerely hope you will recognize our work.**

---

> ### Author Response · Authors · 2024-08-23
> **Hope to Get Your Reply**
>
> Dear Reviewer 3Hpk,
>
> As the deadline is nearing, we wanted to gently follow up on our recent submission. Your feedback is highly valuable to us, and we would appreciate any updates or further guidance you might have regarding our revisions and responses.
>
> Thank you for your time and consideration.

---

### Author Response · Authors · 2024-08-20
**General Comments to All Reviewers**

We thank all reviewers (Reviewer 3Hpk, Reviewer k4id, Reviewer XZhL, Reviewer 53du) for their valuable feedback.

We are encouraged that the reviewers recognized the **novelty** (Reviewer 3Hpk, Reviewer XZhL), **importance** (Reviewer 3Hpk, Reviewer k4id, Reviewer XZhL, Reviewer 53du), and  **significant and valuable contribution** (Reviewer 3Hpk) of our work. Reviewers found that our SafeSora dataset is **the first video generation alignment dataset** (Reviewer 3Hpk, Reviewer k4id) which is **large enough** (Reviewer 53du) to receive **rich and multi-dimention annotations** (Reviewer 3Hpk, Reviewer XZhL, Reviewer 53du); our **analysis and evaluation is thorough and comprehensive** (Reviewer 3Hpk, Reviewer k4id) providing **a solid foundation for future research** (Reviewer 3Hpk, Reviewer XZhL); our paper is **well-written and structured** (Reviewer 3Hpk, Reviewer k4id, Reviewer XZhL, Reviewer 53du).

We addressed all the reviewer comments below and will incorporate them into the revision. The revised version primarily includes the following updates:

1. In Appendix E, add more experimental **comparison of human and multi-modal LLMs preference labeling**. (Reviewer 3Hpk)
2. In Appendix C.2, expand a Prompt Dataset, **specifying source of each prompt**. (Reviewer k4id)
3. Include **the implementation of some traditional metrics** in the dataset and compare them with human preferences. (Reviewer k4id)
4. **Clarify descriptions** in L108 and L152. (Reviewer k4id and Reviewer 53du)
5. In Section 3, **add an example** from the dataset. (Reviewer XZhL)
6. In Appendix D.3, **specify the demographic attributes** of our crowdworkers. (Reviewer XZhL and Reviewer 53du)
7. In Section 5, provide **a more detailed analysis of Figure 6**. (Reviewer 53du)
8. In Instruction section, include more discussion on **the tension between helpfulness and harmlessness in the video domain** to **strengthen the motivation** of the paper. (Reviewer 53du)

If our rebuttal addresses the concerns, we earnestly and kindly ask the reviewers to consider raising the rating and supporting us for acceptance.

---

### Comment · Area_Chair_Mx7b · 2024-08-25

Dear Reviewers,

Thank you for taking the time to review this submission. :)

This is a gentle reminder regarding the reviewer-author discussion.

Please respond to the author's rebuttal at your earliest convenience, especially if you have any points of disagreement.

The deadline is August 31 at 11:59 PM AoE!

Early discussion is always appreciated.

Best,

AC

---

### Comment · Area_Chair_Mx7b · 2024-08-30

Dear Reviewers,

We are just two days away from the end of the discussion period.

Please take a moment to review the author's responses and share any additional feedback you may have.

Best regards,

Your AC

---

### Decision · Program_Chairs · 2024-09-26

**Decision:**

Accept (Poster)

**Comment:**

This work introduces a human preference dataset for text-to-video generation, focusing on the aspects of harmlessness and helpfulness in text-video pairs. The authors collect fine-grained preferences across various subdimensions. I am recommending the acceptance of this submission for the following reasons:

* Unique dataset of text-video pairs with human annotations, which can be useful for future research.
* Insightful analysis on the correlations between different dimensions of human preferences

I believe the paper makes a valuable contribution. However, I encourage the authors to carefully consider how to address the comments, particularly the concerns about annotations raised by Reviewer 53du. It would be beneficial to resolve these issues in the camera-ready version.